# Postoperative Observation of Spaying with the Silicon Ring on the Ovaries in Heifers: A Retrospective Study in 28 Cases

**DOI:** 10.3390/vetsci9110643

**Published:** 2022-11-21

**Authors:** Byung-Hoon Ko, Dong-Gun Park, Won-Jae Lee

**Affiliations:** 1College of Veterinary Medicine, Kyungpook National University, Daegu 41566, Republic of Korea; 2Gogo Large Animal Hospital, Sangju 37267, Republic of Korea; 3Goryeo Large Animal Hospital, Sangju 37213, Republic of Korea; 4Institute of Equine Medicine, Kyungpook National University, Daegu 41566, Republic of Korea

**Keywords:** cattle, heifer, spaying, estrus, postoperative observation, silicon ring, retrospective study

## Abstract

**Simple Summary:**

Because the estrus in cattle is thought to be energy-consuming if the animals are not planned to breed, spaying in cattle has recently been applied to improve daily weight gain and meat quality. However, postoperative changes in ligation-spaying method with the silicon ring to ovaries via transvaginal methods in heifers have not been clearly identified. This retrospective study presented that heifers spayed with this method exhibited no estrus signs at the pubertal age and the ligated ovaries disappeared within a month post-surgery due to ischemic necrosis. Whereas ovarian steroid hormone levels in spayed heifers were not changed following the puberty, luteinizing hormone level at the pubertal age was higher than unspayed controls. Although carcass weight and yield were similar between groups at the pubertal age upon slaughtering, the spayed animals presented higher marbling degree than that of controls. These results may contribute to develop new management strategies for livestock.

**Abstract:**

Although spaying prepubertal heifers has routinely been conducted to control cattle herd and improve meat quality, understandings of the postoperative changes following new spaying methods with the silicon ring on the ovaries via colpotomy remain limited. Therefore, as a retrospective study, 28 cases of spayed heifers were reviewed for postoperative changes after employing this method, with inclusion criteria including complete medical records for clinical observation, ultrasonography, measuring reproductive hormones, and tracking slaughter records. No mortality and heat signs at the pubertal age postoperatively occurred in spayed animals. On ultrasonography during rectal examination, the ovaries were enlarged without any folliculogenesis from one week, while massive ovarian edema appeared from two weeks, and ovaries were no longer palpable at four weeks post-surgery. In hormones, whereas estrogen and progesterone levels did not change from prepubertal to pubertal age in spayed animals, luteinizing hormone levels progressively increased during this period and reached a higher level at pubertal period than unspayed controls. Although carcass weight and yield were similar between groups upon slaughter at pubertal age, the spayed animals presented higher carcass quality (marbling degree) than that of controls. These results may contribute to develop herd management strategies, including control of estrus in cattle.

## 1. Introduction

Cattle are the most economically important livestock in the animal industry. Since they are polyestrus animals, non-pregnant and pubertal females continuously repeat estrus every 21 days. The cattle on the estrus exhibit unique signs, such as mounting behavior (standing to be mounted or attempts to mount another cow), reduced appetite, butting, sniffing, licking, chin-resting, increased activity, and reduced rumination time [1,2]. However, these estrus-related behaviors, which last for approximately 18.4 h every 21 days (3.6% of their life), are thought to be energy-consuming if the cattle are not planned to breed.

The estrus cycle is directly controlled by the central reproductive hormones released from the hypothalamic–pituitary–gonadal (HPG) axis [3]. A pulsatile secretion of gonadotropin-releasing hormone (GnRH) from the hypothalamus is delivered via the hypophyseal portal vein into the pituitary gland where the pulsatile pituitary gonadotropins (follicle-stimulating hormone and luteinizing hormone, FSH and LH) are released. These pulsatile gonadotropins grow the ovarian follicles until the preovulatory follicle stage. Estrogen released from these mature follicles induces the GnRH surge from the hypothalamus, which is followed by ovulation via the LH surge and formation of corpus luteum (CL) in the ovaries. Then, progesterone secreted from the CL negatively feedbacks to the hypothalamus and anterior pituitary gland to reduce the secretion of GnRH and gonadotropins, respectively. Therefore, the removal of the ovaries by spaying represents a fundamental elimination of the source of the gonadal steroid hormone (estrogen and progesterone) in the body, which is followed by breakdown of the HPG feedback system.

Gonadectomy, including spaying (ovariectomy) in females and castration in males, is performed in several species for different purposes. While it is conducted to manage reproduction of companion animals (dogs and cats), spaying in cattle is performed for the purpose of suppressing estrus in feedlot heifers, managing stocking rates, simplifying management, controlling brucellosis, and for removal of diseased ovaries [4,5,6,7]. Because ovarian estrogen is involved in the induction of estrus, spaying in cattle has recently been applied to prevent energy-consuming behavior during estrus period and improve average daily weight gain (ADG) and carcass quality by removing the primary source of estrogen [4,6,8,9]. Given the efforts to secure a more effective and convenient method for spaying in cattle, several surgical methods have been developed using laparotomy as the traditional method, colpotomy for the dropped ovary technique with the aid of a Willis or Kimberling–Rupp (K-R) spay instrument and an endoscope [10,11]. In recent days, the newly developed method of ligation-spaying, involving the application of a silicon ring onto the ovarian hilus via colpotomy, has been applied to spay heifers as a means to induce ischemic necrosis in ovaries.

An accurate understanding of the reproductive physiology of livestock animals is closely associated with the successful development of new management strategies. However, postoperative changes and consequences associated with new spaying method using the silicon ring have not yet been clearly identified. Therefore, the present retrospective study aimed to review postoperative changes following this spaying method in heifers, by clinical observation, transrectal ultrasonography, enzyme-linked immunosorbent assay (ELISA), and tracking carcass reports for carcass yield and quality (marbling degree).

## 2. Materials and Methods

### 2.1. Animals and Inclusion Criteria

Data in the present study were obtained from medical records from 2 large animal hospitals (Goryeo large animal hospital and Gogo large animal hospital, Sangju-si, Gyeongsangbuk-do, Republic of Korea; latitude 36°08′ N and longitude 128°08′ E) for spayed cattle between March 2020 and September 2020, as a retrospective study. From the pool of cattle-spaying cases, the inclusion criteria for selecting the subjects in the present study were as follows: (1) spaying in Hanwoo prepubertal heifers (species: Korean native beef cattle; age: 9.2 ± 0.6 months old); (2) body condition score (BCS) of 2.5–3.5 when animals were selected for spaying; a general consensus of BCS observation was a scale of extremely thin (1) to very fat (5) in the cattle; (3) spaying method with the silicon ring on the ovaries via colpotomy; (4) complete records by veterinarians for clinical observations that were possible to be occurred after the surgical procedure (sudden death, temporary loss of appetite, depression, fever, and astasia) and signs of heat (vulvar edema, clear mucosal vaginal discharge, and/or standing to be mounted) from the pubertal age (approximately 15-months-old); (5) transrectal ultrasonography to ovaries at day 1 (prior to spaying) as well as week 1, week 2, and week 4, postoperatively; (6) ELISA for reproductive hormones including FSH, LH, estrogen, and progesterone; (7) carcass report for carcass yield and quality (marbling degree) at pubertal ages (approximately 30 months of age). After application of the inclusion criteria, 28 cases of spayed cattle were included in the present study from the pool of spaying-related medical records (*n* = 102). As a control group (*n* = 30), medical records from similarly aged but non-operated Hanwoo heifers from the same farms were selected; they were raised with spayed heifers during this study period and slaughtered at similar date. While the postoperative changes (clinical observation, ultrasonography to ovaries, reproductive hormonal profiles, and carcass records) were overall reviewed in spayed group in this retrospective study, the medical records of control group were only involved to present standards (reproductive hormone profiles and carcass records) in intact (unspayed) animals.

### 2.2. Spaying Procedure

The animals were fasted and only permitted to drink water for 12 h prior to surgery. Before surgery, the vulvar and perineal areas were disinfected with iodine-povidone (diluted 1:100). Epidural anesthesia in the sacrococcygeal space was induced using lidocaine hydrochloride injection (0.5 mg/kg; JEIL LIDICAINE INJECTION, JEIL PHARMACEUTICAL, Daegu, Republic of Korea) between the last sacral and first coccygeal vertebrae. To sedate the animals, xylazine (0.05 mg/kg; UNI XYALZINE, UNIBIOTECH, Anyang, Republic of Korea) was intramuscularly injected. The next step was initiated only if complete physical restraint with stanchions and ropes was achieved. The perforator, ligator, and two silicon rings (Sooho-Angel Spay Device, Republic of Korea) were disinfected with 70% ethanol (Appendix A). Then spaying using the silicon ring via the colpotomy was performed. In detail, a perforator (size: 15-mm diameter) was introduced into the vagina and inserted through the vaginal fornix into the caudal abdominal cavity. A perforation was then made by instantly piercing the dorsolateral vaginal wall. After introducing the ligator through the hole with one hand, the ovary was palpated and manually grabbed with another hand during the rectal examination. The grabbed ovary was placed into the hole in the socket of the ligator, and two silicon rings were placed at the ovarian hilus by pushing the shaft of ligator. The same procedure, using a hole in the vagina, was repeated for the other ovary to achieve bilateral spaying. During this procedure, while a veterinarian conducted local anesthetizing, sedating, and spaying of animals, a peer veterinarian helped to monitor the animal’s condition, played an assistant role, and recorded the duration of surgery. After spaying, the enrofloxacin (10 mg/kg; Baytril 100, Bayer, Leverkusen, Germany) was intramuscularly injected once as an antibiotic to prevent possible infections post-surgery. Postoperatively, the animals were carefully monitored for 48 h for sudden death due to internal hemorrhage and other clinical symptoms, such as temporary loss of appetite, depression, fever, and astasia. Moreover, the presence of signs of heat was continuously observed twice a day at feeding times (approximately 07:00 and 17:00) for 10 min until slaughtering day by the veterinarians and/or farmers.

### 2.3. Periodical Ultrasonography in the Postoperative Observation Period

In the spayed group, before initiating spaying at day 1, a transrectal ultrasound scanning of both naïve ovaries was conducted to assess their morphology and measure their major axis, using a portable ultrasound machine and a 5.0 MHz transrectal linear probe (iSCAN, DRAMINSKI, Olsztyn, Poland). In brief, after restraint to animals, the rectum was empty with lubricated and gloved arm. After grabbing the ovary through the rectal wall by hand, the lubricated ultrasound probe was carefully inserted into the rectum and situated near to the ovary to take ultrasound image. This was repeated after 1 week (week 1), 2 weeks (week 2), and 4 weeks (week 4) post-operatively. During this procedure, the number of palpable ovaries via the rectal palpation was counted.

### 2.4. Reproductive Hormonal Profile during Postoperative Observation Period

The serum concentrations of FSH, LH, estrogen, and progesterone were measured in the spayed group and control groups using ELISA. In the spayed groups, at the day of conducting transrectal ultrasound scanning (day 1 preoperatively; 1 week, 2 weeks, and 4 weeks postoperatively) as well as the day before moving to a slaughterhouse, approximately 10 mL of blood was collected from the jugular vein in plain blood collection tubes (Becton Dickinson, Plymouth, UK). The collected blood was allowed to clot for 15 min and then centrifugated at 1000× *g* for 10 min at 4 °C. The isolated serum was stored in a deep freezer at −80 °C until use. The sera collected from control group with same methods before the day of slaughtering were also included in this assay. ELISA kits for measuring reproductive hormones were obtained from Cayman Chemical Company (Ann Arbor, MI, USA). After the stored serum was thawed, a mixture of the sample, enzyme immunoassay (EIA) buffer, tracer, and antiserum was incubated for 60–90 min at room temperature (RT) for estrogen and progesterone level analysis. Thereafter, samples in the 96-well plate were reacted with development reagents (Ellman’s Reagent) for 60 min in an incubator. For FSH and LH, samples, anti-FSH (or LH)-horseradish peroxidase, and anti-FSH (or LH)-biotin-conjugate were mixed and placed into streptavidin-precoated 96-well plate. After incubation for 60 min, the wells were incubated with 3,3`,5,5`-Tetramethylbenzidine (TMB) substrate for 15 min, after which stop solution was added. The reacted 96-well plates were read at 405–450 nm wavelength using a microplate reader (Epoch, Biotek, Winooski, VT, USA). The concentration of each hormone in the serum was calculated using a four-parameter logistic fit with a free software (www.myassay.com: accessed on 2 December 2020).

### 2.5. Carcass Records after Slaughtering

All animals, including those in the spayed and control groups, were slaughtered at approximately 30 months of age, which was a routine slaughtering age in case of beef cattle in Korea. Carcass records after slaughtering animals, issued by a quality evaluator who was employed in the slaughterhouse in compliance with Enforcement Rules of the Livestock Industry Act of Republic of Korea, Chapter 5, Article 38 and evaluated by a quality evaluator in accordance with standards in the Livestock Industry Act of Korea, Chapter 4, Provision 2 of Article 35, were obtained. Then, several values in the carcass records (total carcass weight, carcass yield, and carcass quality) were segregated for further analysis. Carcass yields indicated the percentage of animal weight after slaughter, without blood, viscera, head, legs, tail, kidneys, pelvic fat, and leather, in relation to the live weight before slaughter. Carcass yields were classified as A, B, and C in the related Act of Korea, which were scored as 3, 2, and 1 points in the present study, respectively. Carcass quality (marbling degree), the degree of intramuscular fat in the meat, was rated as 1++, 1+, 1, 2, and 3 in the related Act of Korea, which were scored as 5, 4, 3, 2, and 1 in the present study, respectively.

### 2.6. Statistical Analysis

After confirmation of normality and homocedasticity of obtained values, the values were analyzed using one-way analysis of variance (ANOVA) with Tukey’s post hoc test in case that postoperative data obtained at day 1, week 1, week 2, and week 4 were statistical compared or Student’s *t*-test when statistical assessment between groups (control vs. spayed animal) were conducted (SPSS 12.0, SPSS Inc., Chicago, IL, USA). Differences were considered statistically significant at *p* < 0.05.

## 3. Results

### 3.1. Postoperative Clinical Observation of Spayed Heifers

The results for postoperative clinical observations in spayed animals are shown in Table 1. In the present study, the surgical time was 320 ± 18 s, no animals died, a small number of animals exhibited temporary loss of appetite, and no spayed animals showed signs of heat until pubertal age and slaughtering day, indicating that the ligation-spaying method with the silicon ring is safe for animals and is convenient for surgeons.

### 3.2. Periodical Ultrasonography during the Postoperative Observation Period

The ovaries of spayed heifers were scanned using transrectal ultrasound on day 1 prior to the surgery and at weeks 1, 2, and 4 post-surgery (Figure 1). The normal prepubertal ovary at day 1 had a major axis diameter of 10.2 ± 3.1 mm with small follicles (Figure 1A and Table 2). At week 1, the diameter was markedly enlarged (16.2 ± 4.2 mm) without any indications of folliculogenesis (Figure 1B and Table 2). Massive ovarian edema and significantly more increased swelling (*p* < 0.05; 23.1 ± 7.5 mm) than on day 1 appeared in hypoechoic images at week 2 (Figure 1C and Table 2). As shown in Table 2, most of the ovaries were palpable until week 1 after surgery, but approximately half of them only remained palpable at week 2. At week 4, ovaries were no longer palpable in the appropriate position, indicating that the infarcted and necrotic ovaries had dropped into the abdomen.

### 3.3. Reproductive Hormonal Profile during Postoperative Observation Period

The reproductive hormones (estrogen, progesterone, FSH, and LH) were analyzed by ELISA on day 1, week 1, week 2, week 4, and slaughter date in the spayed heifers (Slau-Spay), as well as the slaughter date in controls (Slau-Con), as shown in Figure 2. The spayed heifers did not present any significant differences in terms of reproductive steroid hormone levels (estrogen and progesterone) until week 4, which was still in the prepubertal period (Figure 2A,B). Although the spayed animals still exhibited lower levels of both hormones at the pubertal period (Slau-Spay), the control animals demonstrated significantly (*p* < 0.05) higher levels at similar ages (Slau-Con). The FSH level tended to increase as postoperative time passed, but this increase was not statistically significant (Figure 2C). The pubertal LH values of the spayed heifers (Slau-Spay) were significantly (*p* < 0.05) higher than those in their prepubertal period and those of the pubertal controls (Figure 2D).

### 3.4. Carcass Records after Slaughtering

The records from the abattoir after slaughtering the animals at approximately 30 months of age were analyzed for carcass weight, yield, and quality, as shown in Figure 3. There was no significant difference in carcass weight or yield between spayed and control heifers, indicating spaying in heifer was not affected to the body weight gain (Figure 3A,B). However, the spayed heifers presented significantly (*p* < 0.05) higher scores in carcass quality (marbling degree) than controls (Figure 3C), which indicated that the meat would be considered more valuable in the Republic of Korea.

## 4. Discussion

Several additives have been developed to suppress estrus in cattle, including melengestrol acetate, anabolics, and deslorelin. However, given concerns about the consumption of beef contaminated with these agents, it is necessary to identify other technical alternatives that can improve fattening systems without the use of compounds that may leave harmful residues in the meat [4,12]. In particular, daily administration of melengestrol acetate, which is widely used in cattle farms, is very important because most heifer’s ovaries initiate to cycle within two or three days if a dose is missed for even one day [13]. Therefore, there is a need for permanent and basic methods for controlling estrus in cattle. To control estrus in cattle and manage stocking rates, simplify management, control brucellosis, use for research purposes, remove diseased ovaries, control mammogenesis, and improve meat quality, spaying in cattle has been widely applied in recent times [10,14]. There are many benefits to spaying heifers in the livestock industry. Because spayed heifers are non-cycling, they can be raised with male cattle in the same pen, which saves stall space and limits unwanted pregnancies. Additives, such as progesterone, are no longer necessary. Furthermore, spayed heifers provide performance benefits as compared to cycling heifers because they do not allocate energy to maintain an estrus cycle or to mount each other during heat. Therefore, since herd management of cattle is the key to improve income as well as reproductive performance, an accurate understanding of their physiology and health status after a veterinary clinical practice is necessary for successful development of new management strategies. The present retrospective study reviewed the postoperative changes and carcass quality associated with ligation-spaying method with the silicon ring via colpotomy for prepubertal heifers. The results showed that the infarcted ovaries were removed within a month and the reproductive hormones (estrogen and progesterone) were reduced, but LH level was elevated in the pubertal period of spayed animals, and spayed heifers produced higher fat-marbled meat. In addition, this method was shown to be safe for animals (rare death and mild side effects after surgery) and convenient for surgeons (high success rate and short surgical duration).

During the present study, none of the spayed heifers presented signs of heat at pubertal age (Table 1). In mares, dropped ovaries can revascularize in the omentum [15]. In addition, revascularization of the ovarian remnant and its hormone production have been reported to occur in eight of nine cats [16]. When the ovarian pedicles of adult dogs were ligated via a ventral midline abdominal approach, the ovarian tissues were changed including severe edema, hemorrhage, and necrosis, but the functions still existed in terms of developing ovarian follicles, CL formation, and response to the external gonadotropins [17]. In cattle, while spaying with laparotomy completely prevented pregnancy, the Willis dropped ovary technique was 92–97% effective when spayed cattle were exposed to bulls, suggesting that pregnancy might occur as a consequence of the ovarian remnants, unless care was taken to ensure removal of the entire ovary [18]. In contrast to the aforementioned surgical method, the ligation-spaying method with the silicon ring in the present study may induce ischemic necrosis in the ovary, which makes revascularization of the dropped ovary impossible.

The increase in body weight and meat quality after spaying cattle remains controversial. No benefits in weight gain were observed in flank-ovariectomized heifers [9]. Moreover, a body weight increase was found in young spayed cows, but not between spayed and control in older cows [4]. Other studies have rather reported a negative effect on body weight gain in spayed cattle with flank laparotomy and colpotomy, as they showed decreased ADG and weight gain:feed ratios that meant an animal’s efficiency for converting nutrients in feed into body mass [9,19]. Of note, when a similar spaying method to the present study (the transvaginal way using a latex ring in the ovarian pedicle) was conducted in heifers, weight gain was lower than controls 56 days post-surgery [20]. However, as opposed to these results, spaying in females could induce body weight gain in several other animal species. With gonadal steroid suppression in rats, spayed females demonstrated a higher increase in weight than in the sham rats, with no significant differences in food consumption, water intake, or animal behavior [12]. It has been reported that up to 68% of spayed dogs become obese possibly due to increased appetite, decreased metabolic rate, and changes in gastrointestinal hormone secretion related to satiety (cholecystokinin and glucagon) [21]. In cattle, spayed young females presented faster weight gain than those that were not spayed [9,22]. Compared to old cattle, spayed young cattle showed improved body weight gain by 60 days after surgery, as well as increased carcass weight [4]. Evidence also suggests that intact heifers have higher circulating estrogen levels than steers and spayed heifers, resulting in tougher beef from intact animals [23,24]. In the present study, the prepubertal spayed heifers monitored until the pubertal age (approximately for 15 months) did not show significant changes in body weight or in carcass yield in comparison with the unsprayed controls; however, carcass quality (marbling degree) was significantly improved after surgery (Figure 3). Further studies are necessary, but the discrepancies among previous reports are thought to be due to dissimilar age at the time of surgery (prepuberty vs. puberty), different breeding management (grazing vs. stall housing; type of rough feed and ratio of concentrate), and different monitoring period (short term vs. long term).

The functions of ovary on growing follicles and secretion of reproductive steroid hormones are controlled by a complex regulatory network with not only well-known reproductive hormones such as gonadotropins but also other cytokines including basic fibroblast growth factor, nerve growth factor, and tumor necrosis factor α in cattle [3,25]. In addition, as the autocrine and/or paracrine factor, estrogen and angiotensin II produced from bovine CL at luteal phase can affect the function of the CL [26,27]. Therefore, removal of ovaries by spaying may induce a breakdown of these complex regulatory networks and feedback system, which is followed by occurring different pattern of hormone balance in the body. In the present study, reproductive steroid hormones in the gonads of spayed animals are clearly decreased. Likewise, the serum levels of estrogen and progesterone in surgically spayed female dogs decreased with a high correlation with time post-operation [28]. No significant differences were found in progesterone levels between spayed cattle and controls at day 0, but the difference was significant at day 90 after surgery [4]. In addition, spaying in female animals induces changes in other hormones, such as gonadotropins, at another level, because the feedback mechanism for gonadal hormones in the HPG system is lacking. More specifically, given that the gonadal steroid hormones (progesterone) provide negative feedback to the hypothalamus and anterior pituitary in order to decrease the secretion of GnRH and LH, there will be no or reduced negative feedback by steroid hormones in the spayed females, which can result in supraphysiological circulating levels of LH in the blood of spayed animals [21]. Likewise, the serum obtained from humans who underwent bilateral ovariectomy with hysterectomy showed that FSH and LH levels increased progressively following surgery [29]. During the first year after spaying in dogs, gonadotropin concentrations significantly increased and then remained at a level around 10 times that of intact dogs [30]. Before spaying (six-month-old), the mean LH concentration values in the serum were similar in both intact and ovariectomized heifers, but secretion of LH was higher in the spayed heifers than in the intact heifers from the age of eight months [8]. In Figure 2, the levels of estrogen and progesterone in the prepubertal period of heifers were not influenced by spaying and were maintained at a lower level until the pubertal period than that in pubertal intact cattle. Levels of both gonadotropins progressively increased following postoperative periods; the elevation of FSH was non-significant, but the LH in the serum was significantly abundant during the postoperative period as well as at pubertal age. Of note, spayed heifers exhibited significantly higher levels of LH at the pubertal age than did their counterparts, possibly due to the absence of negative feedback of progesterone to the hypothalamic–pituitary system.

Traditionally, spaying in cattle has been conducted via flank laparotomy with or without the use of anesthetics or analgesics [12]. Recently, access to the ovaries via the vagina (colpotomy) and dropping ovaries has been achieved by several methods (Willis spaying, K-R spaying, Dutto method, and ligation-spaying with the silicon ring in the present study) [4,10,31]. These colpotomy-based methods are preferable to flank laparotomy in terms of release of fewer stress-related hormones (cortisol, creatine kinase, aspartate aminotransferase concentrate, and haptoglobin), presentation of more normal behaviors (changes in feeding, standing head down, and self-licking), short-term pain, lower morbidity (improved wound healing), and lower mortality in the animals [12]. Of note, because postoperative bleeding during other colpotomy-based methods can easily go unnoticed and can thus be fatal, dropped ovaries via ischemic necrosis achieved by the ligation-spaying method with the silicon ring in the present study is believed to prevent internal bleeding most effectively among several colpotomy-based spaying methods [10]. However, in all colpotomy-based spaying methods, care should be taken to avoid damaging the viscera and uterine vessels by the vaginal perforator, as these surgical techniques are blindly conducted [5]. Therefore, colpotomy must only be performed by a veterinary surgeon experienced in the technique to prevent adverse effects, such as acute death [11,18]. In addition, the use of these methods is limited when the vagina is too narrow for colpotomy or when the subject has large ovarian tumors/adhesions [11]. To overcome these limitations, a new method using an endoscope has recently been introduced for cattle spaying. Laparoscopic spaying offers various advantages with regard to minimal invasion of the abdominal cavity and reduced pain, but it involves costly, specialized instrumentation and time-consuming procedures [10,11,32,33].

## 5. Conclusions

Spaying in cattle has been widely used to control estrus, which is related to improving carcass quality. Therefore, we believe that the present study may contribute to understanding the consequences of the ligation-spaying method with the silicon ring and developing new management strategies for livestock.

## Figures and Tables

**Figure 1 vetsci-09-00643-f001:**
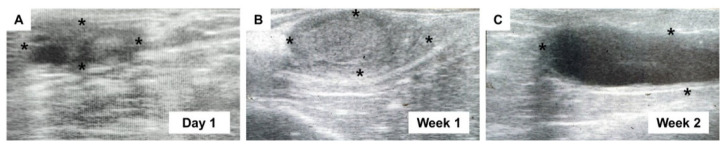
Periodical transrectal ultrasonography at postoperative observation period. Ultrasound scanning to ovaries at day 1 prior to surgery (**A**), and week 1 (**B**) and week 2 (**C**) after surgery. The image for week 4 is unavailable due to dropped ovaries into the abdomen. Asterisks indicate the margin of ovaries.

**Figure 2 vetsci-09-00643-f002:**
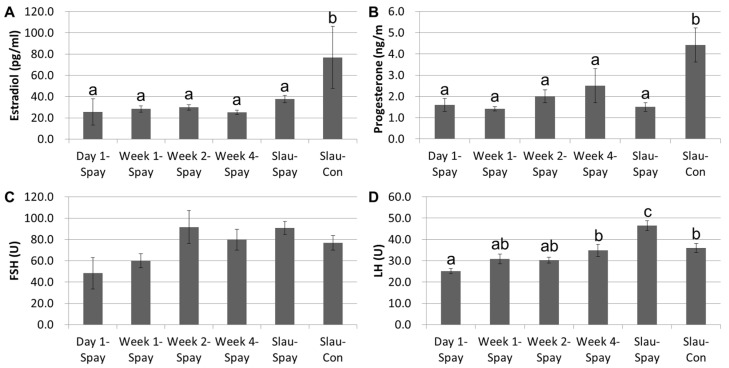
Reproductive hormonal profile during postoperative observation period. (**A**,**B**) The reproductive steroid hormones (estrogen and progesterone) are not significantly changed depending on age and are significantly (*p* < 0.05) lower than in pubertal controls. (**C**) No significant change is observable in FSH level during the postoperative period. (**D**) The pubertal spayed heifers present significantly (*p* < 0.05) greater LH levels than those of their prepubertal period and pubertal controls. Significant differences (*p* < 0.05) are indicated by different letters in the top of bars. Graphs are presented as mean ± SEM. Statistical analysis: one-way analysis of variance (ANOVA) with Tukey’s post hoc test. Abbreviations: FSH, follicle-stimulating hormone; LH, luteinizing hormone; Day 1-Spay, spayed heifers at the day of spaying; Week 1-/Week 2-/Week 4-Spay, spayed heifers after 1 week/2 weeks/4 weeks post-surgery; Slau-Spay, spayed heifers at a day before slaughter day; Slau-Con, unspayed controls at a day before slaughter day.

**Figure 3 vetsci-09-00643-f003:**
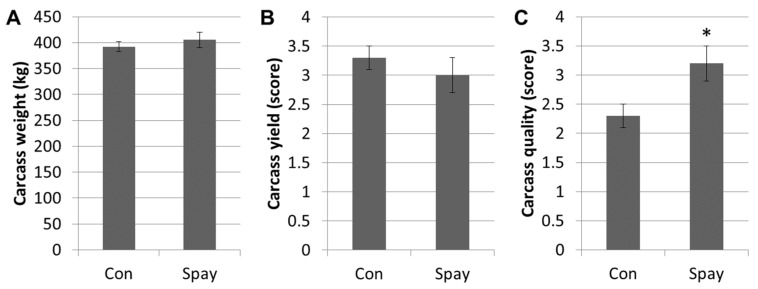
Records for carcass after slaughtering. (**A**,**B**) Both carcass weight and yield are not significantly different between groups. (**C**) Carcass quality (marbling degree) is significantly (*p* < 0.05) greater in the spayed heifers than in controls. Significant difference (*p* < 0.05) is indicated by an asterisk in the top of bar. Graphs are presented as mean ± SEM. Statistical analysis: Student’s t-test. Abbreviations: Con, unspayed control group; Spay, spayed group.

**Table 1 vetsci-09-00643-t001:** Postoperative clinical observation of spayed heifer.

Surgical Time (s)	Mortality	Temporary Loss of Appetite	Depression	Fever	Astasia	Signs of Heat
320 ± 18	0%	14.3%	0%	0%	0%	0%

Surgical time is presented as mean ± standard deviation (SD).

**Table 2 vetsci-09-00643-t002:** Postoperative ultrasonography of spayed heifer.

	Day 1	Week 1	Week 2	Week 4
Number of palpable ovaries	56/56	53/56	25/56	0/56
The major axis of ovary (mm)	10.2 ± 3.1	16.2 ± 4.2	23.1 ± 7.5 *	N/A

Significant differences (*p* < 0.05) is indicated by an asterisk; the number of palpable ovaries is presented as palpable/total ovaries. The major axis of ovary is presented as mean ± standard error of the mean (SEM). Statistical analysis: one-way analysis of variance (ANOVA) with Tukey’s post hoc test. Abbreviation: N/A, not applicable.

## Data Availability

Data will be made accessible from corresponding authors upon request.

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
