# Peer review of "Postoperative Observation of Spaying with the Silicon Ring on the Ovaries in Heifers: A Retrospective Study in 28 Cases"

_vetsci, 2022, doi:10.3390/vetsci9110643_

Round 1

Reviewer 1 Report

The understanding of the reproductive physiology and especially ovary function of cow is closely associated with the successful development of new management strategies. However, spaying in heifers has been routinely conducted, our understanding of the consequences of spaying remains limited. Therefore this is an interesting study demonstrating postoperative observation of spaying with the silicon ring on the ovaries in heifers. The study may contribute in understanding the consequences of the spaying method and developing new management strategies of livestock.

In the introduction the background and the aims of this study are clearly stated. Materials and methods have been described detailed enough. The results are clearly presented and the conclusions are supported by the results. In addition, the data of this manuscript may contribute to develop herd management strategies, including control of estrus in cattle.

Relevant literature has been discussed and cited, however some recent data are omitted. The authors should give some more information about endocrine actions of pituitary hormones, steroids and local-intraovarian factors during different physiological stages of the ovary in context of this study (Please see relevant recent papers of Dr. D. Schams, Dr. A. Miyamoto and Dr. K. Okuda groups).

Author Response

Revision Note for Reviewer #1

 We really appreciate to the precious comments and suggestions from reviewer #1, to improve quality of our work. Based on these comments, the manuscript has been carefully revised. A detailed point-by-point reply to the comments of the reviewers (Revision Note) is provided below for your kind perusal, and any revisions made to the manuscript is marked up using the “Track Changes” function in the revised manuscript.

▪ Comments #1: The understanding of the reproductive physiology and especially ovary function of cow is closely associated with the successful development of new management strategies. However, spaying in heifers has been routinely conducted, our understanding of the consequences of spaying remains limited. Therefore this is an interesting study demonstrating postoperative observation of spaying with the silicon ring on the ovaries in heifers. The study may contribute in understanding the consequences of the spaying method and developing new management strategies of livestock. In the introduction the background and the aims of this study are clearly stated. Materials and methods have been described detailed enough. The results are clearly presented and the conclusions are supported by the results. In addition, the data of this manuscript may contribute to develop herd management strategies, including control of estrus in cattle.

▪ Response #1: We deeply appreciate to the favorable review.

▪ Comments #2: Relevant literature has been discussed and cited, however some recent data are omitted. The authors should give some more information about endocrine actions of pituitary hormones, steroids and local-intraovarian factors during different physiological stages of the ovary in context of this study (Please see relevant recent papers of Dr. D. Schams, Dr. A. Miyamoto and Dr. K. Okuda groups).

▪ Response #2: In agreement with reviewer`s comment, we also think that highlighting that 'the functions of ovary are controlled by a complex regulatory network with a number of hormones and cytokines' is better. Owing to suggestion of reviewer`s suggestion, we haven`t had trouble to find suitable reference articles (research group with Schams, Miyamoto, and Okuda). In revised manuscript, the relevant sentences are placed at lane 360-368 and 500-509.

Reviewer 2 Report

The article talks about “postoperative changes following this spaying method in heifers, by clinical observation, ultrasonography, enzyme-linked immunosorbent assay (ELISA), and tracking slaughter report for carcass yield and quality (marbling degree)”. However, I consider it to be published if some adjustments are made. therefore, I consider it to be accepted with major corrections.

Introduction:

General: I recommend further expanding the section on the problems caused by the manifestation of estrus in cattle and why it is important to implement strategies to counteract it.

Specifics:

Line 48.- If “estrus-related behaviors” is mentioned in the previous paragraph, I consider that it should continue talking about “The estrus cycle” therefore I recommend moving line 63-76 to line 48 and then continue with “Gonadectomy”.

Line 64-76.- cite (authors and year) from where I take these ideas.

Materials and methods:

General: I recommend mentioning the ethics and animal welfare guidelines with which said research was carried out, is there an ethics committee that evaluated said study?

In addition, in results it mentions some variables that it does not describe in this section. Adjust it please.

Specifics:

Line 93.- I consider it important to mention the average age and its standard deviation. In addition to your body condition (mean and standard deviation).

Line 94.- How did they make "records for clinical observation" was it a veterinarian or a technician? Clinical signs were determined according to what criteria?

Line 103.- Why is one mentioned (n=30) and the title mentions 28 cases?

Line 104.- Mentions under what technique the “Ligation-Spaying Method with the Silicon Ring” was performed.

Line 123.- Describe how "transrectal ultrasonography" was performed, that is, the technique performed.

Line 129.- mentions "Investigation" wouldn't it be better to say "hormonal profile"

Line 131.- how did you compare the “spayed group (5 samples) vs control groups (2 samples)?”

Line 149.- mention the technique "Carcass Records after Slaughtering" according to which author

Line 161.- It is necessary to expand the “Statistical Analysis” section. The data was carried out some normality test?, in addition they had homogeneity of variances?. It is necessary to describe the analyzes for other variables, for example “Carcass records after slaughtering”.

Results:

General: Increase the size of the graph literals, they are too small to read.

Specific:

Line 168.- If this section makes a general report of "Postoperative Clinical Observation of Spayed Heifers" and the two experimental groups are not compared, it should go in the materials and methods section and not report it as results (Table 1) or it could be restructured by dividing with data from the control group and spaying group.

Line 169-170.- they mention “After application of the inclusion criteria, 28 spayed animals were included in the present study from the pool of spaying-related medical records (n=102)” because they take data from 102 animals if the research considered only 28 cases?

Line 195.- Table 2, where do you make the statistical comparison for "Number of palpable ovaries"?

Line 213.- for each hormonal value (estradiol, progesterone, FSH and LH) where they report the values ​​of day 1 for the Slau-Spay vs Slau-Con? group. This graph is somewhat confusing, please try to improve it. In addition, the figure "C" lacks literals.

Line 207.- They mention “The FSH level 207 tended to increase with age, but this increase was not statistically significant” and in what section is it mentioned that age is evaluated?

Discussion:

General: I recommend completely restructuring this section. That is, adjust this section according to how I report the results. I recommend discussing what could be the cause of the findings found. For example because they showed a higher (marbling degree) or the difference in hormone levels. More than compare or discuss with data from other species. Also try just to make a discussion according to your results.

Specific:

Line 283.- what does this mean? “ADG and gain:feed (G:F) ratios”

Line 290.- You mention “It has been reported that up to 68% of spayed dogs become obese due to increased appetite, decreased metabolic rate, changes in gastrointestinal hormone secretion related to satiety (cholecystokinin and glucagon), and its influence on the hypothalamus ”, this is more speculative because they do not have values of these hormones to make a comparison. I recommend not to speculate in this section.

Conclusion:

Line 359-361.- this sentence “This retrospective study showed the postoperative changes of the liga-359 tion-spaying method using the silicon ring for dropping ovaries by induction of ischem-360 ic necrosis.” It would be better in the discussion section.

Author Response

Revision Note for Reviewer #2

 We really appreciate to the precious comments and suggestions from reviewer #2, to improve quality of our work. Based on these comments, the manuscript has been carefully revised. A detailed point-by-point reply to the comments of the reviewers (Revision Note) is provided below for your kind perusal, and any revisions made to the manuscript is marked up using the “Track Changes” function in the revised manuscript.

▪ Comments #1: The article talks about “postoperative changes following this spaying method in heifers, by clinical observation, ultrasonography, enzyme-linked immunosorbent assay (ELISA), and tracking slaughter report for carcass yield and quality (marbling degree)”. However, I consider it to be published if some adjustments are made. therefore, I consider it to be accepted with major corrections.

▪ Response #1: We deeply appreciate to reviewer #2`s comments and suggestions to improve quality of our manuscript. From your next comments, we prepared the revision note with point-by-point responses.

▪ Comments #2: (Introduction) I recommend further expanding the section on the problems caused by the manifestation of estrus in cattle and why it is important to implement strategies to counteract it.

▪ Response #2: We already described that the estrus manifestations (mounting behavior, reduced appetite, butting, sniffing, licking, chin-resting, increased activity, and reduced rumination time) are thought be energy consuming if the cattle are not planned to breed or they are raised as beef production (lane 40-47). And it is also noted that spaying cattle has been conducted by several methods to control these estrus behaviors by removing the primary source of estrogen (lane 67-71). Based on the objective of the present study, these contents are considered as sufficient to explain the problems caused by the manifestation of estrus in cattle and strategies to counteract energy-consuming behaviors at estrus. Therefore, we deeply hope that reviewer #2 reconsiders this comment.

▪ Comments #3: (Introduction) Line 48.- If “estrus-related behaviors” is mentioned in the previous paragraph, I consider that it should continue talking about “The estrus cycle” therefore I recommend moving line 63-76 to line 48 and then continue with “Gonadectomy”.

▪ Response #3: In agreement with reviewer`s comment, we also think that switching between these two paragraphs seems better. The relevant correction can be seen at lane 48-77.

▪ Comments #4: (Introduction) Line 64-76.- cite (authors and year) from where I take these ideas.

▪ Response #4: In agreement with reviewer`s comment, this sentence is too subjective to conclude. Therefore, we removed this sentence and corrected the next sentence. The relevant correction is placed at lane 94-97.

▪ Comments #5: (Materials and methods) I recommend mentioning the ethics and animal welfare guidelines with which said research was carried out, is there an ethics committee that evaluated said study?

▪ Response #5: The present study was designed as a 'retrospective study', which means that the data were obtained from the medical records in veterinary clinical practice in the field, were classified under inclusion criteria, and were reviewed as an article. Therefore, because the present study was not designed for prospective study and was not conducted in the academy where the animal research plan must be evaluated by ethics committee in advance, the approval for animal research ethics can be waived. In addition, several retrospective study articles have been normally and routinely published in veterinary sciences without an approval for animal research ethics, in case of retrospective study design. The relevant links are provided below for your kind perusal. Therefore, we deeply hope that reviewer #2 reconsiders this comment.

Relevant links:

https://doi.org/10.3390/vetsci9080420

https://doi.org/10.3390/vetsci9070365

https://doi.org/10.3390/vetsci7040147

▪ Comments #6: (Materials and methods) In addition, in results it mentions some variables that it does not describe in this section. Adjust it please.

▪ Response #6: In accordance with reviewer`s comments #7-15 below, insufficient information was added/revised. Please find response #7-15 below.

▪ Response #7: (Materials and methods) Line 93.- I consider it important to mention the average age and its standard deviation. In addition to your body condition (mean and standard deviation).

▪ Response #7: In accordance with reviewer`s comment, the information is added at lane 108-109.

▪ Comments #8: (Materials and methods) Line 94.- How did they make "records for clinical observation" was it a veterinarian or a technician? Clinical signs were determined according to what criteria?

▪ Response #8: The list for clinical observation consists of general symptoms after surgical procedures, in the field of veterinary clinics. The clinical observation was done by veterinarian who conducted spaying. To make this sentence clear, in accordance with reviewer`s comment, the sentence is revised, which can be seen at lane 110-111 and 149-151.

▪ Comments #9: (Materials and methods) Line 103.- Why is one mentioned (n=30) and the title mentions 28 cases?

▪ Response #9: The spayed cattle (n=28) was compared with unspayed ones (n=30) regarding hormone changes and carcass quality as control; in Korea` law, the animals who will be moved to other places (slaughter house or other farms) must be checked for infectious disease-free using blood. Therefore, this blood used for hormonal assay as control group. Because this retrospective study is aimed to review postoperative changes in spayed cattle who selected by inclusion criteria (n=28) from the pool of cattle-spaying cases (n=102), we used the term '28 cases'. In agreement with reviewer`s comment, '(n=30)' is moved to more proper place. And additional descriptions are added. The relevant correction can be seen at lane 106 and 118-126.

▪ Comments #10: (Materials and methods) Line 104.- Mentions under what technique the “Ligation-Spaying Method with the Silicon Ring” was performed.

▪ Response #10: In agreement with reviewer`s comment, the descriptions are corrected, which is placed at lane 128 and 136-137.

▪ Comments #11: (Materials and methods) Line 123.- Describe how "transrectal ultrasonography" was performed, that is, the technique performed.

▪ Response #11: Because transrectal ultrasonography is very routine clinical practice, we skipped to described the procedure. However, to improve the understanding to whom is not majored in veterinary filed, in agreement with reviewer`s comment, the relevant information is added at lane 156-159.

▪ Comments #12: (Materials and methods) Line 129.- mentions "Investigation" wouldn't it be better to say "hormonal profile"

▪ Response #12: In agreement with reviewer`s comment, the relevant descriptions are corrected, which is placed at lane 162, 245, and 262.

▪ Comments #13: (Materials and methods) Line 131.- how did you compare the “spayed group (5 samples) vs control groups (2 samples)?”

▪ Response #13: In accordance with reviewer`s comment, our description can make the reader confused for blood collection day; in fact, 5 and 2 that you mentioned is not the number of samples but the dates of blood sampling. Therefore, we changed the sentences, which can be seen at lane 120-126, 166-168, and 172-173.

▪ Comments #14: (Materials and methods) Line 149.- mention the technique "Carcass Records after Slaughtering" according to which author

▪ Response #14: The carcass is evaluated by a quality evaluator in accordance with standards in the Livestock Industry Act of Korea, Chapter 4, Provision 2 of Article 35. And the carcass record is issued by a quality evaluator who was employed in the slaughter house in compliance with Enforcement Rules of the Livestock Industry Act of Korea, Chapter 5, Article 38. In accordance with reviewer`s comment, this information was added at lane 190-197 and 200-202.

▪ Comments #15: (Materials and methods) Line 161.- It is necessary to expand the “Statistical Analysis” section. The data was carried out some normality test?, in addition they had homogeneity of variances?. It is necessary to describe the analyzes for other variables, for example “Carcass records after slaughtering”.

▪ Response #15: In agreement with reviewer`s comment, more information regarding statistical analysis is added, which can be seen at lane 205-212.

▪ Comments #16: (Results) Increase the size of the graph literals, they are too small to read.

▪ Response #16: In agreement with reviewer`s comment, the letters in the graph are enlarged at Figure 2 and 3.

▪ Comments #17: (Results) Line 168.- If this section makes a general report of "Postoperative Clinical Observation of Spayed Heifers" and the two experimental groups are not compared, it should go in the materials and methods section and not report it as results (Table 1) or it could be restructured by dividing with data from the control group and spaying group.

▪ Response #17: As reviewer #2 mentioned, the results at "3.1. Postoperative Clinical Observation of Spayed Heifers" and Table 1 are describing the clinical symptoms "only in spayed group". We think that this comment is originated together from your comment #9. Because it is important to show "the ligation-spaying method with the silicon ring is safe for animals" (described at lane 220-221) according to the objective of the present study, it seems to be better that this paragraph is placed here as it is. However, in agreement with reviewer`s comment, the first sentence "After application of the inclusion criteria, 28 spayed animals were included in the present study from the pool of spaying-related medical records (n=102)" is better to move to suitable place at materials and method part. The relevant descriptions lane 215-216 are move to lane 118-120. In addition, the corrected in agreement with reviewer`s comment at lane 120-127 is believed to solve this confusion.

▪ Comments #18: (Results) Line 169-170.- they mention “After application of the inclusion criteria, 28 spayed animals were included in the present study from the pool of spaying-related medical records (n=102)” because they take data from 102 animals if the research considered only 28 cases?

▪ Response #18: As mentioned at response #9, this retrospective study is aimed to review postoperative changes in spayed cattle who are selected by inclusion criteria (n=28) from the pool of all cattle spaying-related medical records (n=102). Like other retrospective studies (linked below), this way for data collection is general and common in the retrospective study; during reviewing data and writing manuscript, the total number of medical records, inclusion criteria for sorting out the reviewable medical records from the pool of total medical records, and the number of reviewed medical records must be mentioned as a retrospective study. Therefore, we here described the total number of medical records for spaying cattle that we had (n=102) and the number of sorted medical records that we reviewed (n=28). However, in agreement with reviewer`s comment, to prevent any confusion about the number of animals in the present retrospective study, we revised the explanation of the number of reviewed animals, as revised at lane 118-126.

Relevant links:

https://doi.org/10.3390/vetsci9080420

https://doi.org/10.3390/vetsci9070365

▪ Comments #19: (Results) Line 195.- Table 2, where do you make the statistical comparison for "Number of palpable ovaries"?

▪ Response #19: In agreement with reviewer`s comment, we missed the method for counting number of palpable ovaries. The relevant description is added at lane 160-161.

▪ Comments #20: (Results) Line 213.- for each hormonal value (estradiol, progesterone, FSH and LH) where they report the values of day 1 for the Slau-Spay vs Slau-Con? group. This graph is somewhat confusing, please try to improve it. In addition, the figure "C" lacks literals.

▪ Response #20: Same with comment #13 and response #13, our description can make the reader confused for blood collection day which is followed by hormonal assay in ELISA. Therefore, via the response #13, we changed the sentences, which can be seen at 120-126, 166-168, 172-173, and 270-273. And the 'C' is added in the relevant place in figure 2.

▪ Comments #21: (Results) Line 207.- They mention “The FSH level tended to increase with age, but this increase was not statistically significant” and in what section is it mentioned that age is evaluated?

▪ Response #21: In agreement with reviewer`s comment, this sentence is inadequate. Therefore, we corrected this sentence, which can be seen at lane 255.

▪ Comments #22: (Discussion) I recommend completely restructuring this section. That is, adjust this section according to how I report the results. I recommend discussing what could be the cause of the findings found. For example because they showed a higher (marbling degree) or the difference in hormone levels. More than compare or discuss with data from other species. Also try just to make a discussion according to your results.

▪ Response #22: In the first paragraph in discussion part, we focused to tell: 1. The advantage of surgical spaying in cattle than additive supplementation to control the estrus in cattle; 2. The summary of the outcome of ligation-spaying method with the silicon ring via colpotomy as a promising method for spaying cattle. In the second paragraph in discussion part, we mainly tell: 1. Adverse effects of other spaying method; 2. Advantage of the ligation-spaying method with the silicon ring. In the third paragraph in discussion part, we extensively reviewed: 1. whether spaying in female can induce body weight gain; 2. Comparing the outcomes for body weight gain between previously published articles and the present study. In the fourth paragraph, we discussed the reason of LH elevation in the present study, with comparing the results from other species. In the fifth paragraph in discussion part, we would like to introduce existent colpotomy-based spaying methods in cattle. Therefore, since we think that our discussion is enough to explain our findings with comparing other relevant published articles, we deeply hope that reviewer #2 reconsiders this comment.

▪ Comments #23: (Discussion) Line 283.- what does this mean? “ADG and gain:feed (G:F) ratios”

▪ Response #23: The average daily weight gain is already abbreviated as ADG at lane 70. In case of gain:feed ration, the detailed explanation was added at lane 336-337.

▪ Comments #24: (Discussion) Line 290.- You mention “It has been reported that up to 68% of spayed dogs become obese due to increased appetite, decreased metabolic rate, changes in gastrointestinal hormone secretion related to satiety (cholecystokinin and glucagon), and its influence on the hypothalamus”, this is more speculative because they do not have values of these hormones to make a comparison. I recommend not to speculate in this section.

▪ Response #24: In agreement with reviewer`s comment, speculative words are removed, which can be seen at lane 346-347.

▪ Comments #25: (Conclusion) Line 359-361.- this sentence “This retrospective study showed the postoperative changes of the ligation-spaying method using the silicon ring for dropping ovaries by induction of ischemic necrosis.” It would be better in the discussion section.

▪ Response #25: In agreement with reviewer`s comment, this sentence is removed.

Reviewer 3 Report

Interesting information, but difficult to read as written.  Inclusion of someone more experienced in writing in the English language is recommended.    A marked up copy with highlights is attached.

Significant editing and improvement to the manuscripts will be needed if the paper is to be published.  

Author Response

Revision Note for Reviewer #3

▪ Comments #1: Interesting information, but difficult to read as written. Inclusion of someone more experienced in writing in the English language is recommended. A marked up copy with highlights is attached. Significant editing and improvement to the manuscripts will be needed if the paper is to be published.

▪ Response #1: We really appreciate to the precious comments and suggestions from reviewer #3, to improve quality of our work. Unfortunately, we cannot find any mark-up or memos in attached file that you uploaded; for double-checking, I captured and attached the downloaded file at next page. Meanwhile, the manuscript has been carefully reviewed by an experienced editor whose first language is English and who specializes in editing papers written by scientists whose native language is not English. I copy the certificate below. In addition, during this 1st revision, we additionally checked the grammar error. Therefore, we would like to ask that reviewer #3 reconsiders this comment.

<Certificate of editing>

<Downloaded PDF>

Reviewer 4 Report

The manuscript presented for review lacks the consent of the ethics committee for this type of research. Has consent been given? If so, it must be completed.

There is no described method of anesthesia of the animals. Were anesthetics administered?

Were post-surgery therapies implemented (Antibiotics, anti-inflammatory drugs)?

The presented form of hormone concentrations is very unclear (Figure 2.). I propose to make separate graphs for each hormone and describe it well. Now you can see estrogen and P4 levels increase after spaying?

Why was it decided to evaluate changes using a 5 MHz probe? Was the Doppler technique used?

Were silent estrus observed in both the control and test groups?

No comparison with another method - traditional ovariectomy.

Was morphology done before and after surgery?

Author Response

Revision Note for Reviewer #4

 We really appreciate to the precious comments and suggestions from reviewer #4, to improve quality of our work. Based on these comments, the manuscript has been carefully revised. A detailed point-by-point reply to the comments of the reviewers (Revision Note) is provided below for your kind perusal, and any revisions made to the manuscript is marked up using the “Track Changes” function in the revised manuscript.

▪ Comments #1: The manuscript presented for review lacks the consent of the ethics committee for this type of research. Has consent been given? If so, it must be completed.

▪ Response #1: The present study was designed as a 'retrospective study', which means that the data were obtained from the medical records in veterinary clinical practice in the field, were classified under inclusion criteria, and were reviewed as an article. Therefore, because the present study was not designed for prospective study and was not conducted in the academy where the animal research plan must be evaluated by ethics committee in advance, the approval for animal research ethics can be waived. In addition, several retrospective study articles have been normally and routinely published in veterinary sciences without approval for animal research ethics, in case of retrospective study design; the relevant links are provided below for your kind perusal. Therefore, we deeply hope that reviewer #4 reconsiders this comment.

Relevant links:

https://doi.org/10.3390/vetsci9080420

https://doi.org/10.3390/vetsci9070365

https://doi.org/10.3390/vetsci7040147

▪ Comments #2: There is no described method of anesthesia of the animals. Were anesthetics administered?

▪ Response #2: We missed the sedation step before colpotomy. The relevant description is added at lane 132-133.

▪ Comments #3: Were post-surgery therapies implemented (Antibiotics, anti-inflammatory drugs)?

▪ Response #3: We missed to describe the use of the antibiotic after surgery. The relevant description is added at lane 145-146.

▪ Comments #4: The presented form of hormone concentrations is very unclear (Figure 2.). I propose to make separate graphs for each hormone and describe it well. Now you can see estrogen and P4 levels increase after spaying?

▪ Response #4: When we visualized the hormone profiles, we agreed that it was better to show results from reproductive hormones all at once in a figure; this might be more convenient to the future readers when they simultaneously compare the increase/decrease of each hormone depending on post-operative period and pre-/post-pubertal period. Therefore, we deeply hope that reviewer #4 reconsiders this comment.

▪ Comments #5: Why was it decided to evaluate changes using a 5 MHz probe? Was the Doppler technique used?

▪ Response #5:   The 5 MHz transrectal probe is widely supplied and used in Korea. Unfortunately, most of popular portable transrectal ultrasound machine are not equipped for Doppler function. Therefore, we could not check the blood flow through the ligated blood vessels at ovarian hilus.

▪ Comments #6: Were silent estrus observed in both the control and test groups?

▪ Response #6: As described in manuscript, signs of heat were monitored only in spayed group. During this procedure we checked whether there is the estrus-related behavior every day as well as growing follicles by rectal palpation (RP) once a week. After 4 weeks post-surgery, because there was no palpable ovary at the female reproductive system, we stopped the RP; but daily observation for the estrus-related behavior was continued. Therefore, we can say that heat in spayed cattle including silent heat is completely monitored. In accordance with reviewer`s comment, the insufficient description is added at lane 149-151.

▪ Comments #7: No comparison with another method - traditional ovariectomy.

▪ Response #7: This retrospective study was aimed to review the post-operative changes in spayed heifers with ligation-spaying method using the silicon ring for dropping ovaries. Therefore, other methods were not considered to review together. To strength the contents of our study, we reviewed other several colpotomy-based methods at discussion part at lane 396-419. Therefore, we deeply hope that reviewer #4 reconsiders this comment.

▪ Comments #8: Was morphology done before and after surgery?

▪ Response #8: We apologize but we are not so sure for which kind of morphology that reviewer means. If reviewer means body condition, there was no observable changes before/after surgery and between spayed/control animals. If reviewer means morphology of ovary, it already described at Figure 1 and Table 2.

Round 2

Reviewer 2 Report

I consider that it will be published only by correcting this section. therefore, I consider it to be accepted with minor corrections.

Materials and methods

Specifics:

Line 92.- what is a normal body condition? I mean, there are several scales to be able to determine, please mention.

A final recommendation for the authors for future revisions would be to add a document where they responded to each question that was made to them in order to have a better understanding of their changes.

Author Response

Revision Note for Reviewer #2

 We really appreciate to the precious comments and suggestions from reviewer #2 during 2nd revision, to improve quality of our work. Based on the comment, the manuscript has been carefully revised. A detailed point-by-point reply to the comments of the reviewers (Revision Note) is provided below for your kind perusal, and any revisions made to the manuscript is marked up using the “Track Changes” function in the revised manuscript.

▪ Comments #1: I consider that it will be published only by correcting this section. therefore, I consider it to be accepted with minor corrections.

▪ Response #1: We deeply appreciate to reviewer #2`s favorable review.

▪ Comments #2:  (Line 92) what is a normal body condition? I mean, there are several scales to be able to determine, please mention.

▪ Response #2: We apologize our misunderstanding for 'body condition' that you mentioned during 1st revision. In agreement of reviewer`s opinion, it was important to recruit heifers with normal body conditions for spaying surgery. Whenever we conducted spaying to heifers, the BCS criteria to select objects to spay was 2.5-3.5, according to a scale of 1-5. In accordance with reviewer`s comment, relevant descriptions are added at lane 96-98.

Reviewer 4 Report

Dear Authors,

Thanks for the answers and explanation.

My biggest concern is still the lack of approval from the ethics committee. A team of (different) operators and the procedure of anesthesia and surgery is not clearly described. In this case, the key question is: would an ethics committee authorize this type of experiments?

Line 113-114: what concentration and dose were used? The name of the drug and the manufacturer are missing.

Line 115: xylazine (0.05 mg/kg) company?

Line 121: 15-mm diameter was then made by instantly piercing the dorsolateral vaginal wall - how was it measured?

Line 127: enrofloxacin (10 mg/kg) - Company?

Line 131-132: No heat detection procedure described (how many times a day, time of day/night, time duration). Was monitoring used for detailed analysis? Based on this, conclusions are later drawn.

Line 182: post-hoc test: analysis of variance tells us whether or not there are differences in the compared means. However, we do not know which groups these differences exist between. Why was TUKEY test chosen? Was the analysis performed with a different test?

▪ Response # 5: A Doppler study would be very valuable and would add to the cognitive value of this work. In my opinion, the only correct frequency for bovine ovarian testing is 7.5 MHz. Nobody compared these two frequencies and diagnostic possibilities (after surgery).

Response #6: I do not agree with this answer. The control group participates in the same way as the test group. It is pointed out when designing a chronic experiment and submitting a proposal to the ethics committee.

Response # 8: Blood test - basic blood count. RBC and WBC would allow for additional control and assessment of the health status of animals. A comparison of the control group and the test group would yield the results.

Author Response

Revision Note for Reviewer #4

 We really appreciate to the precious comments and suggestions from reviewer #4 during 2nd revision, to improve quality of our work. Based on these comments, the manuscript has been carefully revised. A detailed point-by-point reply to the comments of the reviewers (Revision Note) is provided below for your kind perusal, and any revisions made to the manuscript is marked up using the “Track Changes” function in the revised manuscript.

▪ Comments #1: My biggest concern is still the lack of approval from the ethics committee. A team of (different) operators and the procedure of anesthesia and surgery is not clearly described. In this case, the key question is: would an ethics committee authorize this type of experiments?

▪ Response #1:

The present study is a retrospective study that reviews the medical records of cattle-spaying cases. The reasons that approval of animal research ethics can be exempted in the present study are: 1. The committee for animal research ethic (such as IACUC) only reviews the animal research 'plan' as prospective study. The retrospective study can be initiated after completion of data collection. And planning study design prior to conducting study is not for a retrospective study but for a prospective study; 2. The raw medical records in the present study were written by 2 veterinarians who was working in the field of large animal-veterinary clinic (as written lane 90-91) for a living. Because the committee for animal research ethic can review the plan of research group that belongs to same institution, these 2 local animal clinics are out of boundary; 3. The objects for informed consent statement for a study is not for animals but for humans; 4. It is not difficult to find that the retrospective studies review medical records without approving animal research ethics. However, in accordance with reviewer`s comment, we added several information at lane 7-8, 90-91, and 136-138.

▪ Comments #2: Line 113-114: what concentration and dose were used? The name of the drug and the manufacturer are missing.

▪ Response #2: We apologize our mistake. The relevant information is added at lane 121-122.

▪ Comments #3: Line 115: xylazine (0.05 mg/kg) company?

▪ Response #3: We apologize our mistake. The relevant information is added at lane 123-124.

▪ Comments #4: Line 121: 15-mm diameter was then made by instantly piercing the dorsolateral vaginal wall - how was it measured?

▪ Response #4: In agreement with reviewer`s comment, perforation of 15-mm diameter at the vaginal wall was not measurable. Therefore, we change the relevant sentences, which is shown at lane 128-130.

▪ Comments #5: Line 127: enrofloxacin (10 mg/kg) - Company?

▪ Response #5:   We apologize our mistake. The relevant information is added at lane 139.

▪ Comments #6: Line 131-132: No heat detection procedure described (how many times a day, time of day/night, time duration). Was monitoring used for detailed analysis? Based on this, conclusions are later drawn.

▪ Response #6: In accordance with reviewer`s comment, we added the relevant information at lane 144 and 206-207.

▪ Comments #7: Line 182: post-hoc test: analysis of variance tells us whether or not there are differences in the compared means. However, we do not know which groups these differences exist between. Why was TUKEY test chosen? Was the analysis performed with a different test?

▪ Response #7: Because the data exhibited normality and homocedasticity, we could further proceed to use ANOVA to statistically compare in more than 3 groups. Since Tukey`s post hoc is the most widely-used among several post hoc in ANOVA, we chose it. In addition, other post hoc tests such as Bonferroni and Duncan also presented same pattern of significant differences. Following reviewer`s comment, we additionally note the relevant statistical methods at each figures, which can be seen at lane 232-233, 256-257, and 274.

▪ Comments #8: (Response # 5) A Doppler study would be very valuable and would add to the cognitive value of this work. In my opinion, the only correct frequency for bovine ovarian testing is 7.5 MHz. Nobody compared these two frequencies and diagnostic possibilities (after surgery).

▪ Response #8: In agreement with reviewer`s opinion, we also agree that a 7.5-MHz transducer is the better to observe small ovarian structures such as developing follicles, than a 5.0-MHz one. However, it is not difficult to search similar articles with ultrasound scanning to ovaries using a 5.0-MHz transducer. In fact, because we did not have a 7.5-MHz transrectal linear probe, we could not get the images under a 7.5-MHz transducer, and neither under doppler ultrasound. As aforementioned, because the present study is a retrospective study, what we can do is not to prospectively perform the experiment according to planned study design but to review/discuss data from the selected medical records following the objective of study. We are deeply sorry but further study and/or plan-changed study cannot be performed. Therefore, we deeply hope that reviewer #4 reconsiders this comment.

▪ Comments #9: (Response #6) I do not agree with this answer. The control group participates in the same way as the test group. It is pointed out when designing a chronic experiment and submitting a proposal to the ethics committee.

▪ Response #9: As we response at comment #1, approving animal research ethics by committee is not mandatory in the retrospective study.

▪ Comments #10: (Response # 8) Blood test - basic blood count. RBC and WBC would allow for additional control and assessment of the health status of animals. A comparison of the control group and the test group would yield the results.

▪ Response #10: In agreement with reviewer`s comment, checking blood indices from complete blood count and serum chemistry is a good way to monitor the health status of animals. However, as described at lane 94-107, the present retrospective study reviewed the selected medical records which was selected under our inclusion criteria. Because we did not conduct the checking blood indices during postoperative period, the relevant data was not described at our medical records and was not concerned as inclusion criteria. Therefore, we are deeply sorry but further study and/or plan-changed study cannot be performed, and hope that reviewer #4 reconsiders this comment.
